



# The long-standing dilemma of European summer temperatures at the Mid-Holocene and other considerations on learning from the past for the future using a regional climate model

Emmanuele Russo[1,2], Bijan Fallah[3], Patrick Ludwig[4], Melanie Karremann[4], and Christoph C. Raible[1,2]

[1]Climate and Environmental Physics, University of Bern, Sidlerstrasse 5, 3012, Bern, Switzerland.
[2]Oeschger Centre for Climate Change Research, University of Bern, Hochschulstrasse 4, 3012, Bern, Switzerland.
[3]Potsdam Institute for Climate Impact Research (PIK), 14412 Potsdam, Germany.
[4]Institute of Meteorology and Climate Research, Karlsruhe Institute of Technology, Wolfgang-Gaede-Strasse 1, 76131, Karlsruhe, Germany

**Correspondence:** emmanuele.russo@climate.unibe.ch

**Abstract.**

The past as an analogue for the future is one of the main motivations to use climate models for paleoclimate applications. Assessing possible model limitations in simulating past climate changes can lead to an improved understanding and representation of the response of the climate system to changes in the forcing, setting the basis for more reliable information for the

future.

In this study, a Regional Climate Model (RCM) is used for the investigation of the Mid-Holocene (MH, 6000 years ago) European climate, aiming to contribute to end the long-standing debate on the reconstruction of MH summer temperatures for the region, and gaining more insights on the development of appropriate methods for the production of future climate projections.

Two Physically Perturbed Ensembles (PPEs) are first built by perturbing model physics and parameter values, consistently over two periods characterized by different forcing (i.e. the MH and Pre-Industrial (PI) ). The goal is to uncover possible processes associated with the considered changes, that could deliver a response in MH summer temperatures closer to evidence from continental-scale proxy reconstructions. None of the investigated changes in model configuration produces remarkable differences with respect to the mean model behaviour. This indicates a limited sensitivity of the model to changes in the climate

forcing, in terms of its structural uncertainty.

Additional sensitivity tests are further conducted for the MH, by perturbing the model initial soil moisture conditions at the beginning of spring. A strong spatial dependency of summer near surface temperatures on the soil moisture available in spring is evinced from these experiments, with particularly remarkable differences evident over the Balkans and the areas north of the Black Sea. This emphasizes the role of soil-atmosphere interactions as one of the possible drivers of the differences in

proxy-based summer temperature evident between Northern and Southern Europe. A deficiency of the considered land scheme of COSMO-CLM in properly retaining spring soil moisture, evinced from the performed tests and further confirmed by the evidence of present-day studies, suggests that the consideration of more sophisticated schemes may help bridging the gap between models and proxy-reconstructions.





Finally, the distribution of the PPEs with changes in model configuration is analyzed for different variables (T2, PREC, TCLC). In almost all of the considered cases the results show that what is optimal for one period, in terms of a model configuration, is not the best for another characterized by different radiative forcing. These results raise a concern about the usefulness of automatic and objective calibration methods for RCMs, suggesting that a preferable approach is the production of small

PPEs that target a set of model configurations, properly representing climate phenomena characteristic of the target region and that will be likely to contain the best model answer under different forcing.

# 1  Introduction

The Mid-Holocene (MH, approximately 6000 years Before Present (BP)), is one of the main test-beds for evaluating the response of climate models to changes in climate forcing (Otto-Bliesner et al., 2017). For this period, the particular configuration

of the Earth's orbit around the Sun led to important changes in the seasonal cycle of insolation. Knowing whether models react properly to those changes might give us important hints on their reliability for the investigation of the future (Haywood et al., 2019).

The reconstruction of European summer temperatures at the MH has been the subject of a long-standing debate for more than thirty years (Huntley and Prentice, 1988; Cheddadi et al., 1996; Masson et al., 1999; Davis et al., 2003; Mauri et al., 2015;

Russo and Cubasch, 2016). Changes in the seasonal cycle of insolation at different latitudes resulted in a higher summer solar radiation input over the Northern Hemisphere at the MH than today (Berger, 1978; Berger and Loutre, 1991; Berger, 2013). One could expect that climate surface variables, such as near surface air temperature, would have directly responded to the changes in the forcing, with consequently warmer conditions over the entire European continent. Indeed, all climate models, without exception, show a very homogeneous warming in summer across the whole of Europe for the MH, as the result of a

simple direct thermodynamic response to increased summer insolation (Mauri et al., 2015). In contrast, although proxy-based reconstructions are in line with the summer warming shown by models over Northern Europe, evidence from continental-scale proxy-reconstructions show a large extension of colder temperatures over the Mediterranean region at 6000 BP, that is contrary to the results of climate models (Huntley and Prentice, 1988; Cheddadi et al., 1996; Davis et al., 2003; Mauri et al., 2015). Even though the reconstructed MH summer cooling over the entire Mediterranean region has been debated (Samartin et al., 2017),

climate models show a response to the changes in insolation over the entire European continent, with overall simulated warmer conditions, that finds no continental analogue in the proxies (Masson et al., 1999; Mauri et al., 2014; Russo and Cubasch, 2016; Brierley et al., 2020).

Despite different studies have used climate models for investigating MH European summer temperatures, no thorough analysis has been conducted so far on the physical drivers that could plausibly explain the dipole structure evinced from the proxies.

Some studies using the results of climate simulations (such as the results of the Paleoclimate Modelling Intercomparison Project phase 3 (PMIP3)) mainly focused on the average response of the models, rather than investigating whether and for what reason individual models can reproduce European summer temperatures more similar to the proxies (Brewer et al., 2007; Mauri et al., 2014; Samartin et al., 2017). In general, there is a need to better investigate climate models' behaviour for this case study,





exhaustively exploring their physics and testing possible hypotheses that are consistent with the evidence derived from proxy reconstructions.

One of the potential hypotheses proposed in the literature for explaining the cooler summer temperatures over the Mediterranean region at 6000 BP is that more winter and early spring precipitation could have led to increased soil moisture availability

at the beginning of summer. In combination with the enhanced insolation, an increase in latent heat, surface evapotranspiration, and a subsequent decrease of summer surface temperatures can be consequently assumed for the region (Bonfils et al., 2004; Mauri et al., 2015; Russo and Cubasch, 2016). Climate models do not seem to be able to properly partition the incoming radiation between latent and sensible heat, leading to excessive summer temperatures over the area (Russo and Cubasch, 2016). An overestimation of summer temperatures over the Mediterranean region has been noticed also for present-day simulations within

the CORDEX (Coordinated Regional Climate Downscaling Experiment) initiative (Giorgi et al., 2009; Kotlarski et al., 2014). This issue has been related to deficiencies of climate models in properly simulating soil moisture availability at the beginning of summer, resulting from a too fast depletion of spring moisture, and leading to drier and hotter conditions (Seneviratne et al., 2006, 2010; Davin et al., 2016). Similar issues have been also discriminated for soil moisture-controlled evaporative regimes during the Holocene in Central Eurasia (Bartlein et al., 2017). The plausibility of the suggested hypothesis can be effectively

tested with the aid of climate models.

Beside constituting a unique opportunity for better understanding the response of the climate system to changes in the forcing, the study of the MH European climate can be useful also for gaining additional insights on the use of models for the study of the future. Even though climate models are deterministic, changes in their unconstrained model parameter values may lead to different results, assuming a large spectrum of outcomes. The best compromise when applying climate models to study

future or past climate is to calibrate them against observations for the present and determine an optimal model configuration that can be assumed to be the best for other time periods as well (Bellprat et al., 2012a, b; Russo et al., 2019, 2020). However, this is just an assumption, since there is no guarantee on whether the best model configuration for the present will be the same for other periods of time characterized by different forcing. In recent years, so-called objective calibration methods have been developed for tuning both RCMs and GCMs (Bellprat et al., 2016; Hourdin et al., 2017; Hauser et al., 2012; Mauritsen et al.,

2012; Williamson et al., 2015; Bellprat et al., 2012a). In these methods, first a sub-set of changes in model parameters and their mutual combinations are tested. Furthermore, based on these results, a statistical model for extrapolating optimal values of unconstrained parameters is developed. Optimal calibration approaches are based on the performance of a large number of simulations, in a range of 500 to more than 1000 years, being particularly expensive in terms of computational resources. Therefore, their application is not suitable for each case study. In particular, given a continuously increasing complexity of

climate models and the consideration of higher spatial resolutions, their use would put a major constrain on the availability and use of future resources. As an alternative to these approaches, an ensemble of model simulations, referred to as Physically Perturbed Ensemble (PPE) (Forest et al., 2002; Knutti et al., 2002; Murphy et al., 2004; Stainforth et al., 2005; Bellprat et al., 2012a), can be performed to explore different configurations of a specific model for a given period and domain. The advantage of this approach is that beside not making any assumption on the model bias stationarity, it always allows to associate with a

model answer a certain estimate of its structural uncertainty. Investigating how the structural uncertainty of a climate model





varies over two periods of time characterized by different forcing could help to assess the robustness of the assumption of stationarity proper of calibration approaches, and to determine which of the two methods is preferable over the other for the performance of climate projections.

With the main goal of uncovering processes that might be relevant to explain the reconstructed patterns of European summer
temperatures during the MH and the Pre-Industrial (PI) periods, a series of sensitivity experiments are conducted with the regional climate model COSMO-CLM in this study. In a first place, a series of simulations are performed for each of the considered periods by perturbing the model parameters that have proven to be the most sensitive for the region (Bellprat et al., 2012a, 2016) and selecting different physical options that are thought to be relevant for climate feedback related to changes in radiation. Acknowledging the findings of a recent study disputing the usefulness of RCMs for paleo-climate applications
(Armstrong et al., 2019), given that they are mainly limited by biases imposed at their lateral boundaries by the driving GCM, here an RCM is used, mainly because of its relatively cheap computational demands in comparison with high resolution GCMs. This can be undoubtedly considered as an added value of RCMs with respect to high resolution GCMs when aiming to perform a significant amount of sensitivity experiments targeting processes that could have an important impact at local and regional scales.

Secondly, with the aim of supporting the plausibility of soil-atmosphere interactions as the main driver of the bipolar behaviour of summer temperatures over Europe at the MH, another set of sensitivity experiments is conducted with the same model over Europe by perturbing the initial soil moisture conditions of a reference state at the beginning of spring. By evaluating relative changes in summer near surface temperatures of these simulations, this study aims to shed more light on the reasons for climate model biases against proxies.

Finally, in addition to the aforementioned objectives of the paper, the ensembles of simulations with different configurations produced for the study of changes in MH and PI European summer temperatures are used here to assess the reliability of calibration approaches used for RCMs, testing the validity of their assumption of stationarity.

The methods of the study are introduced in Section 2 and include information on the applied model, the performed experiments and the metrics considered in the conducted analyses. In Section 3, the results are presented and discussed. Finally,
conclusive remarks are summarized in Section 4.

## 2 Data and methods

### 2.1 Regional Climate Model

The COnsortium for Small scale MOdelling in Climate Mode (COSMO-CLM, (Rockel et al., 2008)) is a non-hydrostatic, limited-area atmospheric model developed by the Climate Limited-area Modelling-Community (CLM-Community, https://
www.clm-community.eu) an international network of scientists, that join together efforts to develop and use community models (Sørland et al., 2021). COSMO-CLM is the climate version of the numerical weather prediction model COSMO, developed by the German Weather Service (DWD) in the 1990s (Steppeler et al., 2003; Baldauf et al., 2011; Sørland et al., 2021).



The model version used in this study is the COSMO-CLM 5.0_clm9. For its application to study past climates, the model needs to be modified to take into account changes in the Earth's orbit around the Sun on millennial time-scales (i.e. changes in the eccentricity, obliquity and precession). For this purpose a FORTRAN-based subroutine is implemented in the main radiation module of the model code, following the same approach of other paleoclimate studies (Russo and Cubasch, 2016; Prömmel et al., 2013; Fallah et al., 2016, 2018). Additional changes to the model's code are required to account for different greenhouse gas concentrations in the past. An overview of the values of the orbital parameters and greenhouse gas concentrations used in the PI and the MH simulations (based on PMIP3 guidelines, see https://pmip3.lsce.ipsl.fr) is presented in Table 1.

The entire model domain includes 125 grid points in the longitude and 122 grid points in latitude directions, with a spatial resolution of 0.44°($\sim 50$ km), covering entire Europe. At each side of the domain 10 grid points are used as boundary relaxation zone and are excluded from the analysis. A map of the extension and topography of the inner domain of study is presented in Fig. 1.

All performed simulations are derived by applying changes to the setup of a reference run, using an "optimal" configuration slightly different than the most recent one proposed by the CLM-community for Europe (Sørland et al., 2021). The reference run uses the Integrated Forecast System model (IFS) Tiedtke-Bechtold convection scheme (Bechtold et al., 2001) and a 2 time-level Runge-Kutta scheme with time-split treatment of acoustic and gravity waves for time integration. It also uses a second-order Bott scheme for moisture variables and aerosol advection (Bott, 1989) and a prognostic turbulent kinetic energy (TKE) scheme for the vertical turbulent heat and momentum fluxes. Furthermore, a radiative transfer scheme (Ritter and Geleyn, 1992), the multilayer soil model TERRA_LM (Schrodin and Heise, 2002; Schulz et al., 2016)), and a 1-moment, 3-categories (cloud ice, snow, and graupel) ice scheme with prognostic treatment of the hydrometeors (Reinhardt and Seifert, 2006) are applied. The reference run considers 50 atmospheric vertical layers, up to a height of 22000 meters, and a total of 9 hydrological active layers in the soil, down to a depth of 8.62 meters. It also considers a treatment of the albedo based on dry and saturated soil. The main features of the reference simulation are summarized in Table 2.

## 2.2 Driving Global Circulation Model

Initial and boundary data for the COSMO-CLM simulations are obtained from global simulations with the Max-Planck-Institute Earth system model in paleoclimate mode (MPI-ESM-P, (Jungclaus et al., 2013)). Output data with 6-hourly resolution are obtained from the MPI-ESM PI (Jungclaus et al., 2012a) and MH (Jungclaus et al., 2012b) simulations, respectively. The MPI-ESM includes coupled general circulation models for the atmosphere and ocean as well as subsystem models for land/vegetation and for the marine biogeochemistry (Giorgetta et al., 2013). The atmospheric component ECHAM6 (Stevens et al., 2013) is run at T63 horizontal resolution (1.875° on a Gaussian grid) with 47 levels in the vertical.

## 2.3 Physically perturbed ensemble

A total of 31 experiments for each of the considered periods (PI and MH) are performed to build a PPE, leading to a final set of 62 COSMO-CLM simulations. Starting from the reference configuration, selected parameters and physical options are perturbed consistently over the two periods. Most of the perturbed parameters are the ones for which the model has proven



to be most sensitive for Europe (Bellprat et al., 2012a, 2016; Russo et al., 2020), and include at least one member for each of the main model schemes (i.e. turbulence, land-surface, convection, soil and radiation). A set of parameters are considered following the studies of Bellprat et al. (2012a, 2016), and affect sub-grid scale cloud formation (uc1), shallow convection (entr_sc), interaction of radiation with clouds (radfac), turbulent transport of heat and moisture (tkhmin), exchange of heat and

moisture between the atmosphere and the land surface (rlam_heat), strength of transpiration of the vegetation related to depth of rooting zone (facroot_dp2), and hydraulic cycling of soil moisture (soilhyd). Additionally, also the parameters controlling heat exchange between lower atmosphere and ocean (rat_sea), maximal turbulent length scale (tur_len), dissipation of turbulent heat and momentum (d_heat,d_mom), and the factor controlling the effective surface area (e_surf) are considered, based on the sensitivity of the model as evinced from more recent studies (Russo et al., 2020).

The explored physical options of the model are selected considering their potential to be sensitive to changes in radiative forcing, such as the interval of the call to the radiation scheme, the type of albedo representation and the soil hydraulic lower boundary with drainage and diffusion. A list of the different configurations tested starting from the one of the reference simulation are reported in Table 3, together with more detailed information. Each of these simulations covers 25 years, with 5 years considered as spin-up and excluded from the analysis. In total, the performed experiments with changes in the model config-

uration cover more than 1500 years of simulations. The set of conducted experiments is large enough to cover an extensive part of the parameter uncertainty of COSMO-CLM (Bellprat et al., 2012b, 2016), being the main target of objective calibration methods developed in recent years.

### 2.4   Perturbed initial soil moisture experiments

8 additional simulations with perturbed soil moisture conditions are performed over a shorter period of 6 months. All these

20   simulations are initialized on the 1st of April of the 15th year of the MH simulation period and use the same configuration as for the reference experiment of section 2.3. The first of these experiments considers for each point of the domain, each of the hydrological active soil layers as half saturated at inizialization. This means that a value of 50% of relative soil moisture is set for each soil layer at the beginning of the simulation. The relative soil moisture is calculated considering the pore volume of each point of the domain, which is in COSMO-CLM a function of the soil type:

$$\mathrm{W}_{\mathrm{so}_{i,j}} = \frac{W_{l_{i,j}}}{V_{p_{i,j}} \Delta z} \tag{1}$$

where $W_{so}$ is the relative soil moisture for a given point with $x$- and $y$-coordinates $i$ and $j$, respectively, $W_l$ represents liquid water, $z$ the depth of the considered soil layer, and $V_p$ is the pore volume of the considered point (Baur et al., 2018). A total of 8 soil types are available in TERRA_LM.

6 additional simulations are then conducted increasing/decreasing the initial relative soil moisture of the first run by 25%,

30   50% and 75%, respectively. Finally, another experiment is conducted starting from fully saturated soil moisture conditions (+100%) at initialization.





Changes in the mean summer temperature of these experiments are analyzed with respect to the simulation with half-saturated initial soil moisture conditions.

## 2.5 Metrics for evaluating the assumption of stationarity of calibration approaches

The assumption of stationarity proper of calibration approaches is investigated here by means of the Mean Absolute Error (MAE). Three variables that are normally used in RCM calibration procedures are considered (Bellprat et al., 2012a, b; Russo et al., 2020), namely near surface temperature (T2M), precipitation (RR) and total cloud cover (CLCT).

In a first step, the MAE is calculated over daily mean anomalies for each of the 3 variables and each land point of the domain, for the PI and MH periods separately:

$$\mathrm{MAE}^{\mathrm{e}}_{\mathrm{V,i,j}} = \frac{1}{D} \sum_{d}^{D} (X^e_{d,i,j} - X^N_{d,i,j}) \qquad (2)$$

where $e$, $V$, $i$ and $j$ are the selected experiment, the considered variable and the spatial coordinates of a given point, respectively. Additionally, $d$ represents the considered day of the simulation period, with the total number of days $D$ being $20\times365$. Finally, $X^N$ is the target simulation against which the bias is calculated ("nature" state).

In a second step, the MAE is then calculated over regional monthly means, after arbitrarily subdividing the domain of study into a set of 10 sub-regions, similarly to regionalizations usually performed for Europe within the CORDEX framework or in other studies (Kotlarski et al., 2014; Bellprat et al., 2012a, b, 2016). The goal is to mimic the approach used in calibration studies with the same model (Bellprat et al., 2016; Russo et al., 2020). Here, the subdivision is conducted to assign almost each point of the domain to a pre-defined climatic zone. A map of the 10 selected regions is presented in Fig. 2. In this case the MAE is calculated as:

$$\mathrm{MAE}^{\mathrm{e}}_{\mathrm{V}} = \frac{1}{MR} \sum_{m}^{M} \sum_{r}^{R} (X^e_{m,r} - X^N_{m,r}) \qquad (3)$$

where, beside the same indices already introduced in eq. 2, $r$ indicates the sub-region ($R$=10) and $m$ the given month ($M$=12$\times$20).

In both formula for the calculation of the MAE (eq. 2 and eq. 3), one of the model realizations is assumed as representative of the real state of the climate system in the two different periods. All the simulations are then ranked considering their distance from this "nature" realization. The reference simulation of section 2.3 is considered in a first place as the "nature" state. Successively, for supporting the plausibility of the evinced results, the proposed analyses are reiterated using different realizations as the target for the calculation of the MAE.





## 3   Results and discussion

### 3.1   PPE summer temperatures

In this section, PPE results are explored with the main goal of discriminating processes that could lead to a spatial pattern of summer mean near surface temperatures at the MH that is closer to evidence from proxy data. For this purpose, the analyses

focus on the anomalies between the MH and PI climatologies derived from the PPEs with different model configurations. Fig. 3 shows the ensemble mean of the anomalies obtained by subtracting to the MH climatological mean of each realization, the corresponding PI value. The mean anomalies are in a range of +0 to +2.5 °C. The mean model behaviour does not show different results compared to other studies, as the entire domain is characterized by a warming signal. This signal is heterogeneously distributed though, revealing a north-west to south-east gradient, with smaller anomalies over the British

Isles, increasing towards Eastern Europe and the Mediterranean region, and reaching a maximum in the area North of the Black Sea.

The spread of the anomalies of summer temperatures calculated between the corresponding realizations in the two periods is presented in Fig. 4. The spread is given, for each point of the domain, as the difference between the maximum and minimum values of the considered anomalies. The spread is quite constrained, with values exceeding 0.5°C only over parts of the Balkans

and continental Europe. This is the result of a similar structural uncertainty of the model in summer for both periods (see supplements), indicating a limited ability of the model of freely responding to changes in the forcing during the summer season, for any of the considered configurations. Basically, none of the tested model setups produces a considerable different response, with respect to the mean model state, for the two periods. This seems to be true not only for summer temperatures, but also for other seasons and variables (see supplements), pointing at a general stationary of the model uncertainty under

different forcing, at least when considering climatological values. In conclusion, none of the investigated changes in the model configuration, and associated processes, leads to summer temperatures over Europe that could be in better agreement with evidence from continental-scale proxy reconstructions.

### 3.2   Perturbed soil moisture experiments

In this section, the results of the 8 MH experiments with perturbed initial spring soil moisture conditions are presented. Fig. 5

shows the differences in mean summer temperatures of the simulations with changes in initial spring soil moisture (25, 50, 75 % increase/decrease and 100 % increase) with respect to the simulation with half-saturated soil. Changes in available spring soil moisture seem to have an important effect on the simulated summer temperatures of the region (Fig. 5). In particular, there is a strong spatial dependency of the sensitivity of the model to moisture perturbation, with the areas of the Balkans and north of the Black Sea presenting the largest changes, up to 5°C in the case of a reduction of initial soil moisture by 75 %. Differences

in summer temperatures are more pronounced for experiments with initially drier soil, generally presenting warmer conditions (Fig. 5, right column). In contrast, experiments with increased initial soil moisture lead to an overall cooling. In this case though, all the simulations present a very similar spatial distribution of summer temperatures (Fig. 5, left column), independently from the magnitude of the changes applied to the initial conditions. Analyzing the temporal evolution of soil moisture at the different



model levels over the entire 6 months of simulation (Fig. 6), it is evident that the depletion of moisture is faster when more moisture is added to the initial conditions. This leads quickly to similar moisture availability at the beginning of summer in each of the considered experiments, particularly in the upper soil levels. The experiments suggest that even if more soil moisture would be available in early spring in COSMO-CLM, as a consequence of e.g., increased late-winter precipitation, this would

be depleted too quickly, leading to not appreciable changes in summer temperatures. Thus, the use of higher complexity soil schemes, which are able to better retain soil moisture during spring, might have a significant impact on the simulation of MH European summer temperatures, possibly leading to the solution of this complicated puzzle.

This hypothesis is supported by evidences from present-day studies, where a similar warm and dry bias of RCMs against observations over the Mediterranean region is attributed mainly to an overestimation of the evapotranspiration in spring, leading

to a too rapid depletion of soil moisture and, consequently, drier soil conditions in early summer (Seneviratne et al., 2010; Kotlarski et al., 2014; Davin et al., 2016). Davin et al. (2016) solved this issue by coupling the COSMO-CLM to a more complex soil scheme than the default TERRA_LM: the Community Land Model 4.0 (Oleson et al., 2010; Lawrence et al., 2011). In this way, they were able to sensibly reduce the warm summer bias of the model over the Mediterranean region, and confirm the important role of land processes representation to overcome this long-standing deficiency of climate models.

The here presented experiments can not directly explain the disagreement between climate models and proxy reconstructions for mid-Holocene summer temperatures over the Mediterranean region. However, they are particularly important since they confirm that regional differences in European summer temperatures during the Mid-Holocene may be related to soil-atmosphere interactions. In particular, the high sensitivity of large areas of the Mediterranean region to soil moisture perturbation, evident from the presented analyses, supports the idea that there is a strong dependency of near surface temperatures on soil moisture

availability at the beginning of spring for the area.

### 3.3 Testing the assumption of stationarity of calibration approaches for RCMs

In this section, the PPEs produced for the PI and MH periods are used for testing whether an optimal model configuration for one period, can also be assumed to be the best under different forcing. First, the analyses are conducted on daily mean anomalies calculated for each grid point of the domain and for each of the 3 considered variables separately.

The Probability Distribution Functions (PDFs) of total cloud cover daily mean anomalies derived for each member of the ensemble, for a randomly selected point of sub-region 8, are depicted in Fig. 7 as an explanatory example. The PDF of the reference simulation for the PI period is highlighted in black and is considered as a theoretical "nature" state (what would normally be the target of a calibration). In the same figure (Fig. 7, left), the PDF of the PI-experiment with the smallest MAE (eq. 2) with respect to the "nature" state is represented by the red curve. All other ensemble realizations are plotted in light gray.

For the MH (Fig. 7, right), the same colors are used for the same experiments. Thus, the red curve in the MH plot represents again the ensemble member closest to the reference run at the PI period. This aims to show how much the best simulation in one period (PI) diverges from the reference in the other (MH). In each panel of Fig. 7, the number of the best performing experiment for the selected point, in terms of the MAE of eq. 2, is reported in the top-left corner. The optimal realization





changes in the two periods, with simulation 2 (with the exponent to get the effective surface area set to 0.1) being the best in one case, and experiment 26 (with the factor for turbulent heat dissipation set to 15), in the other.

Considering the MAE calculated over the daily mean anomalies (eq. 2) for each land point of the domain, the "best" model configuration changes in the 2 periods for over 91% of the points for 2-meter temperature, 92% for precipitation and 89% for
total cloud cover. The same analyses assuming different realizations as the "nature" state, such as experiment 5 and 9, lead to similar conclusions.

These are further confirmed by additional analyses of the MAE calculated over monthly spatial means, usually being the target of calibration methods employed for COSMO-CLM. Table 4 summarizes the results of the MAE calculated over the regional monthly means of the three considered variables separately, for each of the sub-regions of Fig. 2 together. The sim-
ulations are ranked in Table 4 according to the lowest value of MAE obtained for each variable, from top to bottom. For all of the considered variables, no realization that performs best for one period maintains its "status" in the other. The presented results are not dependent on the considered "nature" state. Again, repeating the same analyses using a different realization as the target (as before, simulation 5 and simulation 9), lead in fact to similar conclusions.

These results suggest that using resources for the calibration of RCMs, in order to determine an optimal model configuration,
might not be the most ideal approach for the study of future and past climate. The production of small PPEs sampling a good part of a model structural uncertainty and likely to contain the best model answer under different forcing would be a preferable option to follow instead.

## 4   Conclusions

In this study, the regional climate model COSMO-CLM is used with the main goal of gaining a better understanding of the
drivers of the long-standing mismatch between outcomes of climate model simulations and proxy reconstructions of Mid-Holocene (MH) summer temperatures over Europe. Additionally, trying to learn from the past for the future, the study also considers MH climate for investigating appropriate approaches for the performance of reliable climate simulations.

Two Physically Perturbed Ensembles (PPEs) are produced to assess how the model reacts, for different parameter values and physical options, to changes in the radiative forcing over two distinct periods (Pre-Industrial (PI) and MH). The mean
differences in seasonal summer temperatures calculated between realizations with the same model configuration for the two considered periods show generally warmer conditions at the MH over entire Europe, consistently with the results of previous modeling studies. In general, each member of the produced PPE does not behave remarkably different with respect to the average model behaviour, for both the MH and PI periods. The spread of the differences between the two periods, calculated for each realization, is in fact very much constrained over most of the domain of study. This indicates a limited sensitivity of the
model to changes in the climate forcing, in terms of its structural uncertainty, suggesting that none of the investigated changes in model configuration, and the associated physical processes, leads to remarkable changes in European summer temperatures closer to evidence of continental-scale proxy reconstructions.





Furthermore, additional sensitivity tests are conducted for the Mid-Holocene by perturbing the model initial soil moisture conditions at the beginning of spring. These experiments show that, for COSMO-CLM, there is a strong spatial dependency of European summer near surface temperatures on the soil moisture available in spring. Remarkable differences are particularly evident over the Balkans and the areas north of the Black Sea, with an increase of up to 5°C when decreasing the initial soil

moisture values by 75%. The differences are more pronounced for the simulations with drier initial conditions, compared to the simulations with enhanced soil moisture. For the latter, similar spatial patterns of colder summer temperatures are evident for all of the considered initial perturbations. Analyses of the temporal evolution of soil moisture show that adding moisture to the initial conditions, leads to a faster depletion, pointing at a deficiency of the considered land scheme of COSMO-CLM in properly retaining spring soil moisture, which has also been confirmed in present-day studies. The conducted sensitivity

experiments emphasize the role of soil-atmosphere interactions as one of the possible drivers of the differences in proxy-based summer temperatures evident between Northern and Southern Europe. The consideration of more sophisticated soil schemes may bridge the gap between models and proxy-reconstructions, contributing to the solution of this long-standing dilemma.

Finally, the analysis of the distribution of the PPEs for different variables (T2, PREC, TCLC) shows that, in almost all of the considered cases, an optimal model configuration in one period does not seem to be the best in another characterized by different

radiative forcing. The ranking (based on the mean absolute error) of the single realizations changes each time. These results raise a concern about the usefulness of automatic and objective calibration methods for RCMs. Since there is no guarantee that an optimal model configuration maintain its status over different periods of time, it might make sense to better channel the use of computational resources. An effective use of resources is of fundamental importance for the production of climate projections and should be considered as one of the main priorities of future climate studies. The presented results suggest that

a better approach to the calibration of RCMs is the production of small PPEs that target a set of model configurations, properly representing climate phenomena characteristic of the target region and that will be likely to contain the best model answer under different forcing.

*Code and data availability.*

Simulations configuration files can be downloaded from:

https://doi.org/10.5281/zenodo.5140094

All the data for the Mid-Holocene period on which the presented analyses are conducted are available at the following link: https://doi.org/10.5281/zenodo.5138131

All the data for the Pre-Industrial period on which the presented analyses are conducted are available at the following link: https://doi.org/10.5281/zenodo.5140034

Additional data used for the performance of the presented simulations, such as land/surface parameters, as well as the interpolated boundaries with soil moisture conditions at 50% saturation, are available at the following link:

https://doi.org/10.5281/zenodo.5140079

The R scripts used for conducting the presented analyses are available at:





https://doi.org/10.5281/zenodo.5144973

A complete documentation of the COSMO-Model is permanently available at the following link:

https://www.dwd.de/EN/ourservices/cosmo_documentation/cosmo_documentation.html

The COSMO-CLM model is completely free of charge for all research applications. The version of the COSMO-CLM model

used in this study can be downloaded from the following website:

https://redc.clm-community.eu/projects/cclm-sp/wiki/Downloads.

Access is license-restricted (http://www.cosmo-model.org/content/consortium/licencing.htm) and for the download the user

needs to become a member of the CLM-Community, or the respective institute needs to hold an institutional license.

*Author contributions.*  ER designed the study and performed the simulations. All authors contributed to the interpretation of the results, the

writing, and scientific discussion.

*Competing interests.*  The authors declare no competing interests.

*Acknowledgements.*  CCR was supported by the Swiss National Science Foundation (SNF) within the project 'PleistoCEP' (grant: 200020_172745).
PL is supported by the Helmholtz Climate Initiative REKLIM (regional climate change; https://www.reklim.de/en). Data is locally stored on
the oschgerstore provided by the Oeschger Center for Climate Change Research (OCCR).

The computational resources necessary for conducting the experiments presented in this research were made available by the German
Climate Computing Center (DKRZ).

The authors are also particularly grateful to the COSMO and the CLM-Community for all their efforts in developing the COSMO-CLM
model and making its code available.





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

**Figure 1.** Map of the topography and extension of the model domain.







**Figure 2.** Domain decomposition.





**Figure 3.** Mean of the anomalies of summer (JJA) mean near surface temperature calculated between each of the ensemble realizations, subtracting to the climatological value of the Mid-Holocene (MH) the one obtained for the Pre-Industrial period (PI).

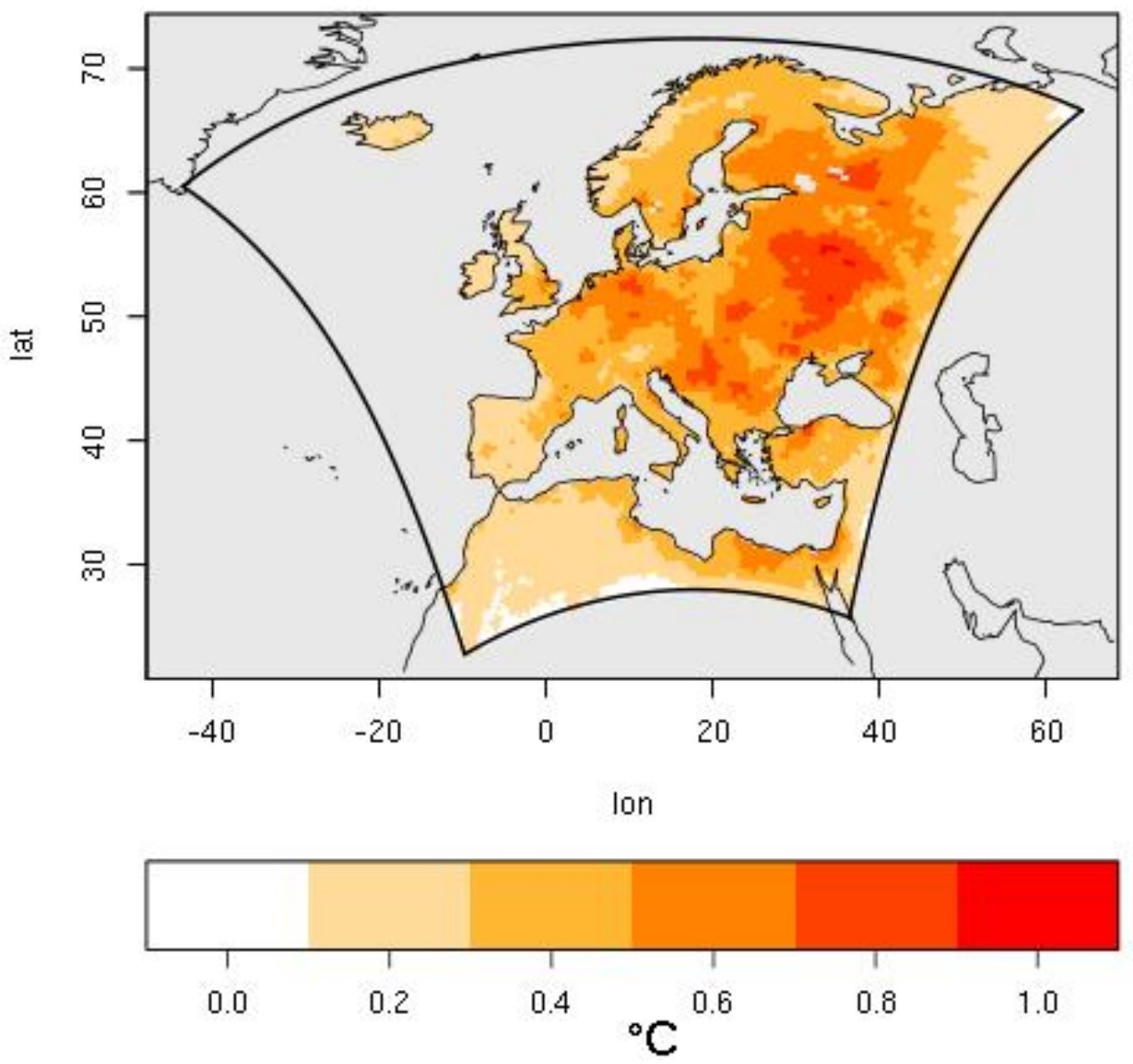

**Figure 4.** Spread of the anomalies of summer (JJA) mea near surface temperature calculated between each of the ensemble realizations, subtracting the climatological value of the corresponding Mid-Holocene (MH) and Pre-Industrial simulations (PI).



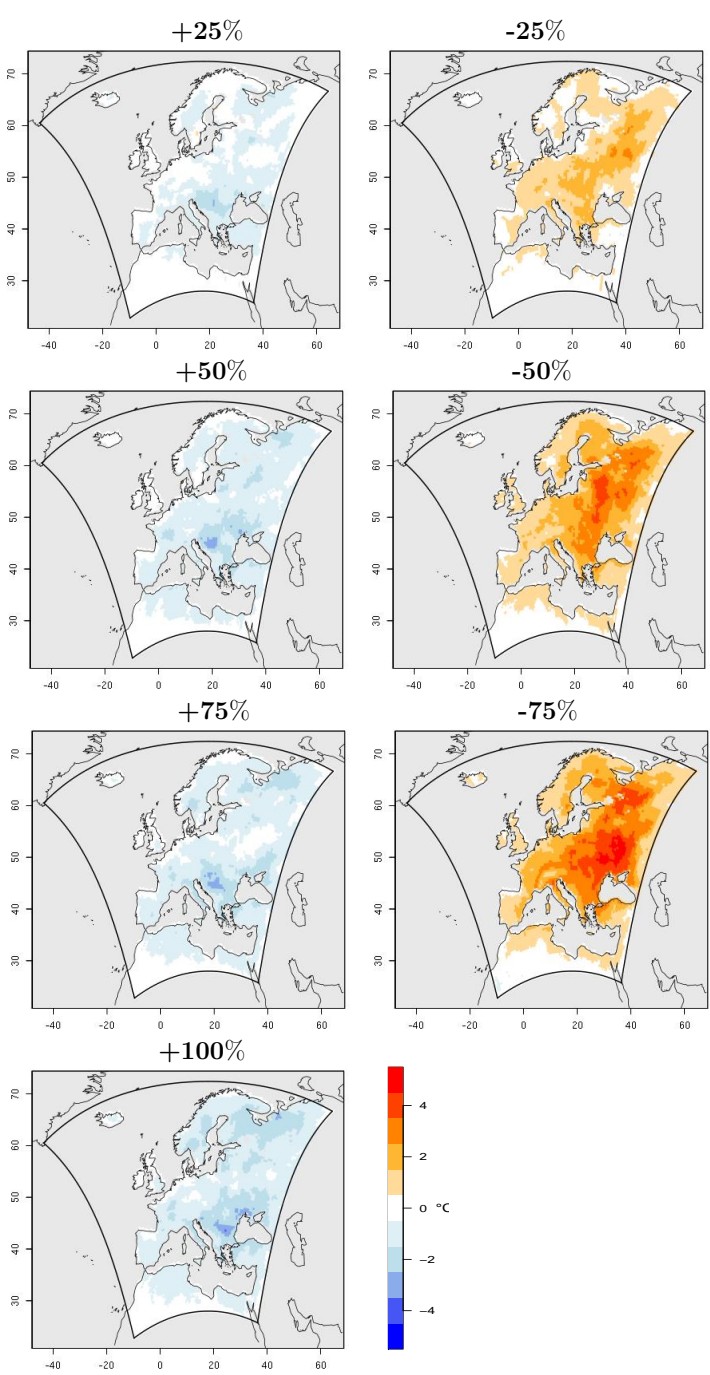

**Figure 5.** Differences in summer (JJA) mean near surface temperature calculated between each of the perturbed soil moisture experiments and the simulation with initial half-saturated soil levels. The left column shows the results of an increase in initial spring soil moisture by 25%, 50%, 75% and 100% from top to botttom, respectively. The right column shows the results obtained with drier initial soil moisture conditions, by 25%, 50% and 75%.







**Figure 6.** Temporal evolution of soil moisture at the nine hydrological active layers (lev1 to lev9) for the experiments with increased initial soil moisture at the beginning of April. The data are shown from the first time step until the end of October of the same year of the simulations.The different experiments with added soil moisture and the reference run (REF: 50% of relative soil moisture for all the points of the domain on the 1st of April) are indicated by different colors.



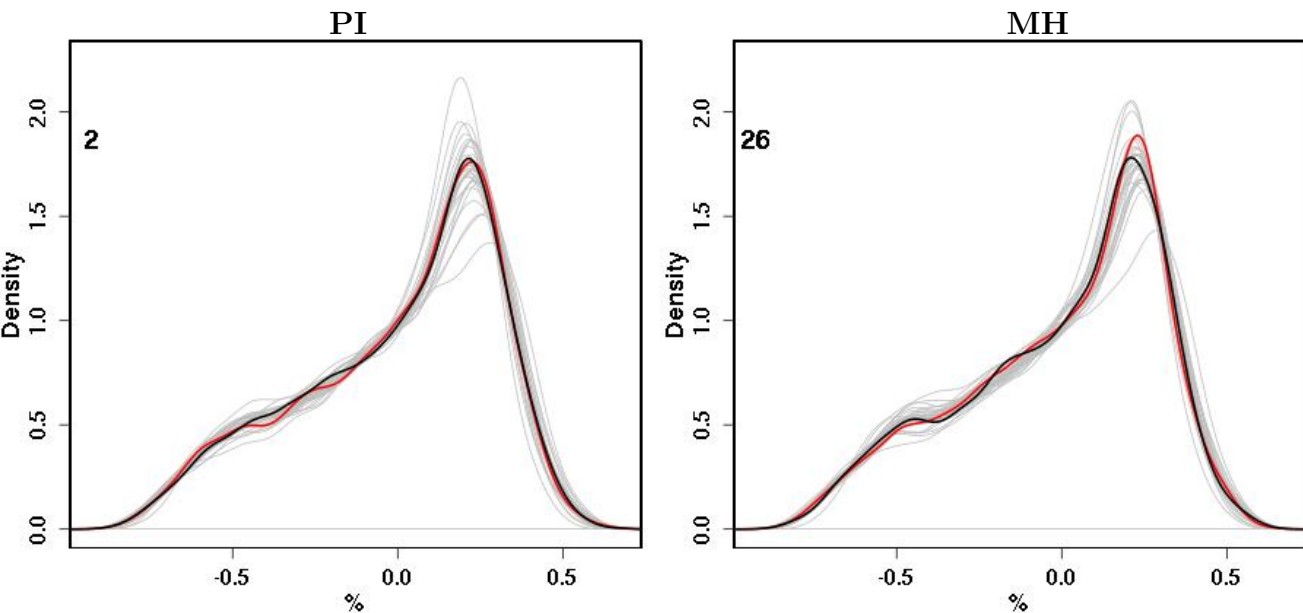

**Figure 7.** Probability Distribution Functions of daily mean anomalies of total cloud cover calculated for the different ensemble realizations for the PI (*left*) and the MH (*right*) periods, for a randomly selected grid point in subregion 8 of Fig. 2. The PDF of the reference run, considered as the "nature" state, is highlighted in black. The realization with the minimum MAE with respect to the PI reference run is highlighted in red in both panels. Gray lines represent the remaining members of the ensembles for the 2 periods.





**Table 1.** Values of Orbital Parameters and Greenhouse Gases Concentrations for the PI (left) and MH (right) periods. The orbital parameters are the Eccentricity of the orbit (ECC), the Obliquity of the Earth's axis (OBL) and the precession of the equinoxes (PRE).

|          | PI        | MH        |
|----------|-----------|-----------|
| **N2O**  | 270 ppb   | 270 ppb   |
| **CO2**  | 280 ppm   | 280 ppm   |
| **CH4**  | 760       | 650 ppb   |
| **ECC**  | 0.016724  | 0.018682  |
| **OBL**  | 23.446°   | 24.105°   |
| **PRE**  | 282.04°   | 180.87 °  |



**Table 2.** General description of model setup for the reference simulation

| | |
|---|---|
| **Spatial Resolution** | $\approx 0.44°$ |
| **Timestep** | 240 s |
| **Convection** | Tiedke-Bechtold |
| **Time Integration** | Runge-Kutta, |
| **Lateral Relaxation Layer** | 500 km |
| **Soil Model** | TERRA-ML SVAT |
| **Albedo** | Surface albedo determined by two external fields for dry and for saturated soil |
| **Rayleigh Damping Layer (rdheight)** | 11 km |
| **Soil Active Layers** | 9 |
| **Active Soil Depth** | 8.62 m |
| **Atmospheric Vertical Layers** | 50 |



**Table 3.** List of experiments performed by perturbing the values of the corresponding parameters and physical options of the reference simulation. The values used in the considered experiments and the ones employed in the reference run are reported on the last two columns of the table.

| Exp. Number | Parameter | Description | Val. | Val. Ref. |
|---|---|---|---|---|
| **1** | **itype_hybound** | Hydraulic lower boundary with drainage (1) and drainage+diffusion (3) | (3) | (1) |
| **3** | **itype_hydbound** | Hydraulic lower boundary with drainage (1) and drainage+diffusion (3) | (3, | (1) |
| | **itype_wcld** | Cloud water diagnosis based on rel. hum. (1) or statistical scheme (2). | 1, | (2) |
| | **imode_tran,** | Diagnostic (1) or Prognostic (2) TKE-equation in transfer scheme | 2, | (1) |
| **29** | **ninc_rad** | Interval (in time steps) between two calls of the radiation scheme | (6) | (12) |
| **30** | **itype_albedo** | Vegetation albedo is modified considering forest fraction. | (4) | (1) |
| **Turbulence** | | | | |
| **8,9** | **tkhmin** | minimal diffusion coefficients for heat | (0,1,2) | (0.35) |
| **27,28** | **tur_len** | maximal turbulent length scale | (100,1000) | (500) |
| **25,26** | **d_heat** | factor for turbulent heat dissipation | (12,15) | (10.1) |
| **23,24** | **d_mom** | factor for turbulent momentum dissipation | (12,15) | (16.6) |
| **Land Surface** | | | | |
| **4,5** | **rlam_heat** | scaling factor of the laminar boudary layer for heat | (0.1,10) | (0.5249) |
| **6,7** | **rat_sea** | ratio of laminar scaling factors for heat over sea and land | (50,100) | (20) |
| **2** | **e_surf** | exponent to get the effective surface area | (0.1) | (1) |
| **Convection** | | | | |
| **19,20** | **entr_sc** | mean entrainment rate for shallow convection | (5e-5, 1e-4, | (3e-3) |
| **21,22** | | | 1e-3, 2e-3) | |
| **Radiation** | | | | |
| **14,15,** | **uc1** | parameter for computing amount of cloud cover | (0.2,0.5, | (0.0626) |
| **16** | | in saturated conditions | 0.625) | |
| **12,13** | **radfac** | fraction of cloud water/ice used in radiation scheme | (0.3,0.9) | (0.5) |
| **Soil** | | | | |
| **10,11** | **soilhyd** | multipl. factor for hydraulic conductivity and diffusivity | (3,6) | (1) |
| **17,18** | **fac_rootdp2** | Uniform factor for the root depth field | (0.5,1.5) | (1) |



**Table 4.** Ranking of different realizations in the two periods (PI left and MH right) for each of the considered variables, based on the MAE calculated over regional monthly means with respect to the reference simulation of Table 2. The simulations are ordered, from top (best) to bottom (worst), based on their MAE rankings (eq. 3)

| T2M | | RR | | CLCT | |
|---|---|---|---|---|---|
| **PI** | **MH** | **PI** | **MH** | **PI** | **MH** |
| 14 | 29 | 25 | 27 | 25 | 29 |
| 25 | 14 | 1 | 13 | 1 | 1 |
| 20 | 21 | 31 | 20 | 26 | 18 |
| 1 | 20 | 14 | 24 | 31 | 17 |
| 21 | 19 | 26 | 23 | 10 | 25 |
| 29 | 22 | 15 | 19 | 29 | 10 |
| 22 | 25 | 30 | 18 | 17 | 26 |
| 19 | 18 | 29 | 30 | 11 | 31 |
| 15 | 15 | 21 | 11 | 24 | 11 |
| 18 | 12 | 11 | 10 | 18 | 24 |
| 10 | 10 | 12 | 28 | 30 | 3 |
| 26 | 1 | 10 | 21 | 14 | 30 |
| 3 | 24 | 18 | 16 | 3 | 14 |
| 24 | 26 | 8 | 25 | 21 | 21 |
| 12 | 3 | 28 | 14 | 28 | 28 |
| 11 | 16 | 22 | 31 | 27 | 27 |
| 17 | 31 | 16 | 2 | 22 | 22 |
| 16 | 13 | 17 | 29 | 2 | 2 |
| 31 | 17 | 13 | 6 | 23 | 23 |
| 13 | 11 | 24 | 3 | 20 | 6 |
| 28 | 28 | 20 | 15 | 6 | 20 |
| 27 | 30 | 3 | 22 | 19 | 12 |
| 30 | 27 | 19 | 26 | 12 | 19 |
| 8 | 8 | 2 | 1 | 4 | 15 |
| 23 | 23 | 27 | 17 | 15 | 4 |
| 4 | 4 | 9 | 12 | 8 | 8 |
| 2 | 2 | 23 | 8 | 13 | 13 |
| 6 | 6 | 4 | 9 | 7 | 7 |
| 9 | 9 | 6 | 4 | 16 | 16 |
| 7 | 7 | 7 | 7 | 9 | 9 |
| 5 | 5 | 5 | 5 | 5 | 5 |