# Peer review of "The long-standing dilemma of European summer temperatures at the Mid-Holocene and other considerations on learning from the past for the future using a regional climate model"

_Climate of the Past, 2021_

## Author Comment (AC1)

Reply to
**1st Reviewer**
*Russo, E., Fallah, B., Ludwig, P., Karremann, M., and Raible, C.C.:*
*The long-standing dilemma of European summer temperatures at the*
*Mid-Holocene and other considerations on learning from the past for*
*the future using a regional climate model*

Dear referee,

thank you very much for accepting to review our manuscript and for the time you dedicated to its revision.

Below we go point by point through your technical corrections, presented in *italic*, detailing how we dealt with your concerns reported in **bold**.

Sincerely,

Emmanuele Russo

**General Comments**

*The authors use a regional climate model to investigate the role of spring soil moisture in influencing summer temperatures over Southern Europe and Mediterranean during the mid-Holocene. The authors find that increasing soil moisture generates cooler summer temperatures, identifying a potential source of model bias that may help explain proxy-based paleoclimate reconstructions that show cooler than present mid-Holocene summer temperatures in many parts of the region while models show a uniform warming. The paper is very well written and the project is well designed. I think that it is eminently suitable for publication in Climate of the Past, and I can thoroughly recommend its publication with only minor changes. The paper provides what I think is an interesting and important contribution to both modern and palaeo climate science. I have some questions and general comments, as well as a few minor technical corrections.*

- *Q1. What influence could change in soil depth and quality have compared to winter rainfall on soil moisture content? I presume that models use modern soils, but if mid-Holocene soils were better quality and*

*depth then presumably they could create a similar effect since it would allow increased retention of winter/spring rainfall. Model soil hydrology is quite crude (especially in GCM's) but there is also quite a considerable body of evidence that suggests the Mediterranean region lost soil in the late Holocene as a result of natural and anthropogenic aridification. This could mean that better soils in the MH could result in more soil moisture being held in the spring, irrespective of any change in winter rainfall. See for instance https://iopscience.iop.org/article/10.1088/1755-1315/9/1/012011. It may be worth adding a comment on this.*

Uncertainty related to changes in soil composition or quality on millennial time scales is something that definitely needs to be better considered when performing and interpreting paleoclimate modeling simulations. Different and better soil would very likely impact the results (as evident also for present-day studies [Guillod et al., 2013, Smiatek et al., 2016]) and is something that deserves to be properly taken into account in future studies for the MH, considering that soil might have plausibly changed on millennial time scales. Our experiments do not allow to exhaustively assess the influence of winter precipitation versus changes in soil composition and quality on soil moisture content available at the beginning of summer. For this task a more comprehensive set of experiments should be performed using a larger ensemble of climate models of different complexity. Nevertheless, following the referee's comment, we will better discuss this point, acknowledging its importance, in the new version of the manuscript.

- *Q2. How does increased soil moisture generate the observed summer cooling? It would be interesting to know to what extent this is a result of, for instance, latent heat, evapotranspiration, clouds or atmospheric circulation changes. Perhaps the authors could add a paragraph on this as it would be interesting to know the degree to which the effects are felt locally (similar to the thermodynamic effect of orbital changes in insolation) or over some distance.*

The summer cooling obtained for the experiments with enhanced spring soil moisture is due mainly to a larger partition of the incoming energy towards latent heat, with a consequent increase in

surface evapotranspiration and near surface temperatures. This is visible from the plots we provide here in Fig. 1. The plots present summer biases in evapotranspiration and latent heat calculated between the simulation with saturated soil (+100%) and the one with 50% relative soil moisture in spring (reference state). The pattern of the two maps is almost the same, and pretty similar to the corresponding map of the bias in near surface temperatures (Fig. 5 of the former version of the manuscript). An excess of latent heat flux (negative sign taken in the upward direction) and evapotranspiration is evident over a large part of the domain, consistent with the pattern of cooler temperatures for the corresponding experiment with enhanced spring soil moisture. Following the referee's comment, we will try to better describe the effect of increased spring soil moisture onto summer temperatures in our experiments, in the new version of the manuscript. However, we would like to avoid having an additional section in the paper, since we think it results already quite lengthy.

- *Q3. The authors mention the debate about summer cooling over southern Europe and the Mediterranean during the mid-Holocene. Their experiments show cooling with increased soil moisture, but is this cooling of sufficient magnitude to override the strong warming in the model and therefore cause the negative temperature anomalies shown in the proxy evidence? As far as I understand it, figure 5 shows the effect of soil moisture on summer temperatures relative to the normal model state at the MH, and not summer temperatures as an anomaly compared to the PI. It would be useful to include a comment or figure (even in the supplementary) on this to see whether it is likely to approach the cooler than present summer temperatures shown in the proxy evidence.*

We want to emphasize here again that the main goal of our study is not to properly quantify the effect of soil moisture on summer temperatures. Rather, we want to show that there is a strong spatial dependency of MH summer temperatures on the soil moisture available in spring over Europe, that needs to be carefully acknowledged when interpreting climate models results. Our experiments are very helpful in this sense: even though the "default" outputs of our model are in agreement with previous modeling works, and also with proxy-based reconstructions such as the one of Samartin

[Figure]

Figure 1: **Differences in summer mean of daily values of evapotranspiration (*left*) and daily mean latent heat flux (*right*), calculated between the simulation with saturated spring soil moisture and the one with 50% relative soil moisture. The sign of the fluxes is taken negative in the upwards direction.**

et al. [2017], its results change remarkably over specific areas when perturbing spring soil moisture. At the same time, following the referee's comment, we want to specify here that figure 5 shows the effect of soil moisture on summer temperatures relative not to the normal model state, but to a state with 50% relative soil moisture in spring. In this case, the summer cooling resulting from increased spring soil moisture seems quite restrained in order to fit the picture drawn from pollen-based reconstructions. However, when considering the differences between the wettest (+100%) and the driest (-75%) of our set of sensitivity tests presented in the former version of the manuscript, we see that very large differences (down to -7°) are evident in particular over the Balkans and the areas North of the Black Sea (Fig. 2 of the current document). This pronounced differences suggest that, over some regions, higher spring soil moisture content could very plausibly help approaching the cooler than present summer temperatures shown in the pollen-based reconstructions. Following the referee's comment, we will review our results and discussion section in the new version of the manuscript, trying to make these evidence clearer to the reader.

- *Q4. P4 10-14; The authors highlight the importance of the GCM in which regional models are embedded (eg Armstrong et al 2019). To what degree could the choice and performance of the GCM impact the result? For instance, we know that GCM's have difficulty simulating the mid-Holocene African Monsoon, and therefore probably the Hadley Cell and sub-tropical high pressure over the Mediterranean in summer. This may be related to my Q2, and particularly to what degree the spatial pattern of cooling caused by soil moisture changes could be dependent on the GCM outside of the regional model (e.g. atmospheric dynamics etc). Maybe a comment would be useful just to say whether this is/is not important, and why.*

The selection of the driving GCM definitely has an important impact on the results of an RCM, especially for Europe [Sørland et al., 2021]. A different spatial sensitivity of European summer temperatures from the soil moisture available in spring/late winter could likely result from the use of a different GCM. This could happen, for example, as a consequence of changes in cloud cover

[Figure]

Figure 2: **Differences in summer mean near surface temperatures calculated between the simulation with saturated spring soil moisture and the one with a 75% soil moisture reduction with respect to the reference with half-saturated soil.**

associated with different large-scale features imposed by the driving GCM, affecting the pattern of incoming shortwave radiation at the surface. We agree with the author that this is a point deserving more attention, and we will try to add some comments in this regard in the new version of the manuscript.

- *P5 22-23; The soil in the model is an important part of the story here. Where has the soil data come from that is used in the model? And what are the main variables used? eg carbon content, particle size, permeability etc. There are different sources with different qualities (eg FAO, EU etc)*

We agree with the referee that detailed information on soil characteristics are missing in the former version of the manuscript. Our simulations use a soil map derived from the digital soil map of the World (FAO, 2003). The soil model has 8 different soil types. For each grid box, the soil in the column belongs to the same soil type. Each soil type has constant values prescribed in the model, for different parameters such as pore volume, field capacity, permanent wilting point, heat capacity, etc. A table with the values of the different soil parameters is provided in Doms et al. [2013]. Following the referee's comment, we will provide more detailed information on the soil map used in our study and on TERRA_LM, together with corresponding references, in the new version of the manuscript.

- *P9 18 P11 11-12; See also my earlier comments in Q1 about MH soils in the Mediterranean region being different than the modern soils in the region*

Please refer to our answer to your previous point.

**Minor technical corrections**

*The text has some minor grammatical errors and typos. I highlight some here, but please take time to have another careful read of the text, particularly from section 3 onwards.*

- *P2 28 'Despite different studies have used..' Different studies have used climate models for investigating MH summer temperatures, but no thorough..*

**Thanks. We will modify this part accordingly.**

- *P4 2 'stationarity proper of..' stationarity in calibration (?)*

**We refer here to the stationarity of the relationships between model outputs and "reality". We will try to make it clearer in the text.**

- *P4 6 'In a first place..' Firstly,*

**We will modify the text accordingly.**

- *P5 9; 'covering entire Europe' covering the whole of Europe*

**Thanks. We will correct this part.**

- *P5 9; 'used as boundary' used as a boundary*

**We agree and will modify this part of the text accordingly.**

- *P9 5; 'to not appreciable..' to no appreciable..*

**Correct.**

- *P9 15; 'The here presented..' The experiments presented here..'*

**We agree and will modify the text accordingly.**

- *P9 23; 'different forcing.' different forcings.*

We will modify the text accordingly.

- *P9 27, P10 5, P10 12; 'nature' do you not mean 'natural' state?*

Here we simply wanted to name the simulation that we assume closer to reality as the "nature" state. Therefore, we used a name instead of an adjective. We feel that this can be well regarded as a personal preference.

- *P9 27-28; 'what would normally' that would normally*

We agree and will modify the text accordingly.

- *P11 17; 'maintain its' maintains its..*

This will be corrected following the referee's suggestion.

**Figures**

- *Fig 1; Scale needs attention, blank above 2500m*

We will remove white from the colorbar.

- *Fig. 3 'subtracting to the climatological...' not sure what is meant here so no suggested replacement text, but the whole sentence needs another look.*

We will revise the caption of Fig. 3, trying to make it easier to read.

- *Fig. 3; Convention would suggest using blue for cooler and red for warmer (use green/brown for drier)*

**Agree and we will change the employed colors for this plot, following the referee's suggestion.**

- *Fig 4; 'mea'? not sure what this means.*

**Here we wanted to refer to the "mean". We will correct this typo in the new version of the manuscript.**

- *Fig 6; Can you use a different value on the x axis rather than hours? I have no concept of how long 1000's of hours are (having looked it up, 1000 hours = 42 days). The y axis would also be better scaled in mm rather than in metres, and it would be easier to understand if the labels for each of the 9 levels included their depths, or at least something to give them more meaning if possible.*

**We will modify the plots of Fig. 6 following the referee's comments.**

**References**

G. Doms, J. Föster, E. Heise, H.J. Herzog, D. Mironov, M. Raschendorfer, T. Reinhardt, B. Ritter, R. Schrodin, J.P. Schulz, and G. Vogel. A description of the nonhydrostatic regional cosmo-model - part ii: Physical parameterizations. Technical report, COSMO - Consortium for Small-Scale Modelling, 2013.

B.P. Guillod, E.L. Davin, C. Kündig, G. Smiatek, and S.I. Seneviratne. Impact of soil map specifications for european climate simulations. *Climate dynamics*, 40(1-2):123–141, 2013.

S. Samartin, O. Heiri, F. Joos, H. Renssen, J. Franke, S. Brönnimann, and W. Tinner. Warm mediterranean mid-holocene summers inferred from fossil midge assemblages. *Nature geoscience*, 10(3):207–212, 2017.

G. Smiatek, J. Helmert, and E.M. Gerstner. Impact of land use and soil data specifications on cosmo-clm simulations in the cordex-med area. *Meteorol. Z*, 25:215–230, 2016.

S.L. Sørland, R. Brogli, P.K. Pothapakula, E. Russo, J. Van de Walle, B. Ahrens, I. Anders, E. Bucchignani, E.L. Davin, M.E. Demory, et al. Cosmo-clm regional climate simulations in the coordinated regional climate downscaling experiment (cordex) framework: a review. *Geoscientific Model Development*, 14(8):5125–5154, 2021.

---

## Author Comment (AC2)

Reply to
**2nd Reviewer**
*Russo, E., Fallah, B., Ludwig, P., Karremann, M., and Raible, C.C.:
The long-standing dilemma of European summer temperatures at the
Mid-Holocene and other considerations on learning from the past for
the future using a regional climate model*

Dear referee,

thank you very much for accepting to review our manuscript and for the
time you dedicated to its revision.

Below we go point by point through your technical corrections, presented in
*italic*, detailing how we dealt with your concerns reported in **Bold**.

Sincerely,

Emmanuele Russo

**General Comments**

*This paper presents the results of a set of simulations performed with the
COSMO-CLM limited area model, with a focus on Europe and the Mid-
Holocene (MH) climate. The main objective is to shed light on the question
why climate models produce warmer summer conditions in Southern Europe
relative to the preindustrial (PI), in contrast with pollen-based reconstruc-
tions that suggest cooler conditions. This is a highly relevant issue in (pa-
leo)climatology, as it is important to know to what extent climate models can
reproduce climate's sensitivity to a change in radiative forcings. The first
step taken by Russo et al. was to investigate the sensitivity of the results to
different model configurations. For this purpose, 30 experiments were per-
formed for PI and MH with different parameter values and perturbed model
physics (so 60 in total). The results of these experiments were rather sim-
ilar, suggesting limited sensitivity. Therefore, it is unlikely that the noted
model-data mismatch is related to the set of parameters used. As a next step,
short experiments were performed for the MH with prescribed different soil
moisture contents in spring. As expected, more humid soils produced lower
surface temperatures during summer, in closer agreement with proxy-based
reconstructions for Southern Europe. The authors suggest that possibly the*

*use of more complex soil schemes in models could result a longer retention of soil moisture during summer and thus lower summer temperatures. A final analysis concerns an evaluation of the model performance in the 60 different experiments, showing that the optimal performance is different for different variables and periods. This highlights that a model setup that is performing well for the present day, is not necessary producing an optimal performance for the past or the future.*

*I expect these results to be of interest for paleoclimate modelers, making this potentially a useful contribution to the literature. However, in my opinion, the manuscript requires a substantial revision before it can be accepted for publication. My main concerns are the following:*

- *A more balanced discussion of the MH summer temperatures in Southern Europe is required. The study assumes that the pollen-based reconstructions are correct, but it is important to make clearer to the reader that there are other proxies that agree with what the models show.*

Here we agree with the referee that more information should be provided about what other proxy-based reconstructions show for European summer temperatures at the MH. It is true that from the way we structured the text it might result to the reader that we think that pollen-based reconstructions are the only correct answer. We aim to revise the text accordingly, trying to provide a more balanced overview of the status of research on the topic. At the same time, though, we believe that the same attention should be paid in the literature to the fact that models have different answers. For this we take a chance here to make it clear that our goal is not to discuss which proxy-based reconstructions are more reliable. On the contrary, playing with a climate model physics and configuration, we want to test the plausibility of some of the hypotheses that were proposed in the literature for explaining the picture drawn from pollen-based reconstructions. These could work in principle in both ways, for either supporting or denying the evidence derived from them. This is a quite an important point deserving more attention, since model uncertainties are not always exhaustively considered in a paleoclimate context. We believe that our manuscript could be helpful in this sense.

- *A few additional experiments with a more sophisticated soil scheme should be performed to make the conclusion of Section 3.2 much more convincing.*

We do not totally agree with this referee's comment. We think that our current experiments already provide an important contribution to the research on the topic. We therefore decided not to perform additional experiments in this regard. Acknowledging that this is a very important point, though, we provide a very detailed and robust motivation for our choice in the corresponding answer to the specific comments of the referee below. At the same time, following the referee's comment, we realized that our conclusions should be carefully reviewed in case there are too strong statements that we cannot exhaustively demonstrate, given the outcomes of the presented experiments.

- *Not all figures and tables provide additional value to the text and could thus be omitted, especially Figure 6 and 7, and Table 4. A section should be added to part 3 to discuss the results against earlier modelling studies on the MH climate in Europe.*

We address the comments to each individual figure and table more in detail in the specific comments section. We agree that figure 6 could probably be moved to the supporting material section, as well as table 4, and we will consider to do so in the next version of the manuscript, if opportune. However, we really like and think that Figure 7 should be part of the main text. Beside acknowledging the possibility of expanding the discussion on the results against previous MH modeling studies, we do not think that an additional section would add value to the presented analyses. We will motivate this choice, providing more details on the way we aim to address this comment as exhaustively as possible in the specific comments section below.

*Further details are provided below.*

- *P. 1, line 6. I propose already mentioning here what RCM is used in this study.*

**We will correct this point accordingly to the referee's comment.**

- *P. 2, line 1-2. Please explain the different variables (T2, PRE, TCLC)*

**We will remove the acronyms of the variables in the abstract, simply saying that we consider different variables.**

- *P. 2, line 21. Please note that only pollen-based reconstructions suggest cooler summers in S Europe, and there is a good explanation why, see for instance Samartin et al. (2017).*

**As said above, following the referee's comments, we realized that we need to make it clearer to the reader that there are other proxies that show something different. The text should be modified accordingly.**

- *P. 2, line 23. Here references to other climate modelling studies should be included. For instance, Strandberg et al. (2014).*

**We will add more references here, including the one proposed by the referee.**

- *P. 2, line 26. ". . . . overall simulated warmer conditions, that finds no continental analogue in the proxies". This is incorrect. The cooler summer conditions at MH are primarily suggested by pollen-based reconstructions. Other proxies (e.g., chironomids, glacier records) do provide evidence for warmer summer conditions during the MH (e.g., Samartin et al., 2017). This point should be discussed more clearly here. It is understandable that the extensive temperature reconstruction of Mauri et al. (2015) is used to evaluate climate models, but it is important to realize that there are also proxies that suggest a contrasting result for summer conditions during the MH. So, there is an alternative explanation to the model-data mismatch, i.e., that the models are correct and the reconstructions are wrong.*

**We agree with the referee on the fact that this period should be reformulated, making clearer that the picture is highly debated in**

the proxy community. At the same time, though, we take a chance here to emphasize again the importance of realizing that models also suffer from uncertainties and they are far from correct. This is true for the present day and, very likely, also for the past. We hope that our results could give an important contribution in this sense, highlighting the need of carefully taking into consideration possible models' misbehaviour.

- *P. 2, line 29. Please clarify the dipole mentioned here.*

Here we refer to the dipole structure of summer temperatures over Europe derived from pollen-based reconstructions. We will make this clearer in the new version of the manuscript.

- *P. 3, line 9. Please provide more information on the overestimation of the summer temperatures. Overestimated by how much?*

The overestimation changes for different models and areas. In some cases it exceeds 5°C [Russo and Cubasch, 2016]. We will add such information in the new version of the manuscript. We will add here also references for additional studies highlighting RCMs common warm biases for summer temperatures over the Mediterranean region such as: Christensen et al. [2008] and Boberg and Christensen [2012] . Finally, we will also add references for CMIP5 and newly available CMIP6 experiments with GCMs, usually showing a similar warm bias in summer temperatures over the Mediterranean region for the present day [Carvalho et al., 2021, Cattiaux et al., 2013].

- *P. 3, line 14. Is this hypothesis really tested in this manuscript? I would rather say "evaluated".*

We agree and will correct the text accordingly.

- *P. 3, line 22. "However, this is just an assumption, since there is no guarantee on whether the best model configuration for the present will*

*be the same for other periods of time characterized by different forcing.” In addition, different parameter sets can produce similar present-day mean climates in agreement with observations, but with different sensitivities to radiative forcing perturbations. See for instance Loutre et al. (2011, Clim. Past 7, 511-526). I propose to mention this here.*

**We agree and will mention this point in the new version of the manuscript, adding relevant references on the subject, including the one proposed by the referee.**

- *P. 3, line 24. Please explain RCM and GCM.*

**We will provide the full explanation for the two acronyms here.**

- *P. 4, line 2. Please clarify what “two methods” you mean.*

**The methods we refer to are calibration versus the generation of Physically Perturbed Ensembles (PPEs). Following the referee’s comment, we will modify this sentence in the new version of the manuscript, trying to make the distinction between the two methods clearer.**

- *P. 4, line 6: I suggest providing references and an explanation of the COSMO-CLM acronym where it is first mentioned, so here, instead of in Section 2.1*

**We agree and will modify the text accordingly.**

- *P. 4, line 22. I suggest mentioning here the calibration approaches to be assessed.*

**Here we wanted to refer to calibration approaches in a general sense and not to a specific method. Following the referee’s comment, we propose to review this part of text in the new version of the manuscript. We will try to clarify here that our goal is to test the stationarity assumption proper of calibration methods used for regional climate models.**

- *P. 4, line 9. The sentence starting with "Acknowledging the findings..." complex and hard to read. Consider revising.*

**We agree with the referee and will revise this part of the text accordingly.**

- *P. 5, line 3. Please note that although obliquity is an important astronomical parameter, it does not describe changes in the Earth's orbit around the sun as suggested in the manuscript.*

**We agree and propose to modify this part accordingly in the new version of the manuscript.**

- *P. 5, Section 2.2. I propose to mention information (including the resolution) on the ocean model here as well. In addition, I suggest mentioning here that the same values for astronomical parameters and greenhouse levels are used in the driving GCM and COSMO-CLM.*

**We agree and will include all the missing information highlighted by the referee about the configuration of the GCM in the new version of the manuscript.**

- *P. 5, line 32. On what is "the reference configuration" based? Please explain.*

**We actually provided very detailed information on the reference simulation at the end of subsection 2.1 of the former version of the manuscript. We preferred to have the description of the reference simulation in subsection 2.1 rather than in 2.3, since this is the reference (the baseline from which all other experiments are conducted) for all the experiments proposed in the different subsections. Therefore, we think that it would be a very legitimate choice to keep the same structure for the description of the reference simulation in the new version of the manuscript.**

- *P. 5, line 31. The text mentions 31 experiments, but Table 3 shows 30 different experiments. Is the 31st experiment the reference run?*

**Yes. We will try to make this clearer in the text.**

- *P. 6, line 19. I do not really see the rationale behind the experiments described in Sections 2.4 and 3.2. As discussed on page 9 (line 10), it is already known for a few years that, when using the default set up, COSMO-CLM has problems retaining the spring soil moisture and that this results in dry soil conditions in summer and anomalously high surface temperatures. So, there is really no need to show that again here. Davin et al. (2016) solved this by applying a more sophisticated soil scheme that performed much better. So instead of the experiments discussed in Sections 2.4 and 3.2 I would suggest evaluating the more complex soil scheme of Davin et al. (2016) for the MH climate. I therefore propose that the authors perform additional experiments with MH conditions and the soil scheme of Davin et al. (2016) and to show the results in figures that replace current Figure 6.*

**The experiments of section 3.2 are not aimed at showing that COSMO-CLM has problems in retaining spring soil moisture, as the referee suggests. The main goal of the proposed experiments is to test the plausibility of the hypothesis that soil-atmosphere interactions could be responsible for the bipolar behaviour of summer temperatures over the Mediterranean region at the MH, as evinced from pollen-based reconstructions, eventually shading more light on the possible reasons for model biases. This hypothesis has been often suggested in the literature, without being effectively tested in practice. In this context, we believe that the presented experiments are very useful, confirming that there is a strong sensitivity of summer temperatures, in our model, from the soil moisture available in spring, in particular over areas such as the Balkans and North of the Black Sea. For these areas we can affirm with our results that more attention should be paid when comparing model results against proxy-based reconstructions. In fact, even though our general results (Fig. 3 and Fig. 4 of the former version of the manuscript) fit very well with the results of previous modeling studies, and they would be in total agreement with the outcomes of some of the most recent study on the subject [Samartin et al., 2017], the same model presents very disparate outcomes as a result of perturbed spring soil moisture, leaving space to different interpretations. This is an important outcome**

of our work, based on the presented experiments, that must be properly acknowledged in future studies on the subject. As for COSMO-CLM, the mentioned bias in summer temperatures over Southern Europe characterizes a large set of regional and global climate models for the present day. The reason for this bias is often suggested to be related to a poor representation of soil processes in the model [Christensen et al., 2008, Boberg and Christensen, 2012, Carvalho et al., 2021, Cattiaux et al., 2013, Kotlarski et al., 2014, Russo and Cubasch, 2016]. Nevertheless, this is something never carefully considered in Mid-Holocene studies for Europe such as the one of Samartin et al. [2017], where the models are considered as generally "correct". We are sure that our experiments could give an important contribution in this sense, highlighting the need of properly considering the skills of the soil component of climate models applied to the study of European MH climate, when interpreting their results. Concerning the comment on the need to show COSMO-CLM misbehaviour in depleting moisture in the soil too quickly, we thought that it was important to provide an explanation to the reader on why, when reducing soil moisture, there is a clear linear trend of the response in summer temperatures, while in the case of wetter soils the response looks very similar in all of the considered cases. These analyses are important in our opinion, since they demonstrate that more complex soil schemes are necessary for the study of MH summer temperatures. Or, more precisely posed, we show that models with low skills in retaining spring soil moisture should be dismissed when investigating summer temperatures at the MH over Europe. In fact, even if a model would present wetter soil in spring, as a consequence, for example, of enhanced late-winter/early-spring precipitation over southern Europe at the MH, its effects on summer temperatures would be lost if the given model had low skills in retaining soil moisture during spring, such as in the case of COSMO-CLM using TERRA_LM. This is something that was never acknowledged before in the literature and represents a fundamental milestone for future and previous studies, again giving weight to our experiments. Since large differences are expected in near surface temperatures depending on soil moisture conditions in spring, as suggested by the referee some lines above, we hope that our results could set the basis for new investigations with models including more complex soil-schemes. For all these reasons we think that

our work deserves publication and no additional experiments using different models are needed. At the same time, we believe that the sensitivity of climate models to soil-moisture perturbation needs to be better assessed, but in a larger context, making use of a plethora of climate models of different complexity. This is beyond the objectives of our paper. This said, we also have to acknowledge that it might not be true that simply considering more complex soil models would lead to the solution of the problem, as we stated in the former version of the manuscript. Even if we demonstrate that soil-atmosphere interactions could have a large impact on summer temperatures over Europe, this would surely not be the only condition for a good match with pollen-based reconstructions: as seen some lines above, a model producing wetter late-winter/early-spring conditions would still be needed. We realize, following the referee's comment, that our conclusions are in this sense too strong and that they need to be revised accordingly.

- *P*. 6, line 28. Please provide a bit more information on TERRA_LM. What are the 8 soil types for example, and how do they differ? How is the moisture holding capacity of these soils established? Are the soil types and their characteristics fixed during this study?

This was a point also highlighted by the first reviewer. The soil classes of TERRA_LM are: ice, rock, sand, sandy loam, loam, loamy clay, clay and peat. The model is uni-dimensional and the soil type for a grid box is the same over all considered vertical layers. 15 different soil parameters such as the pore volume, field capacity and the plant wilting point are fixed for each soil category and prescribed to the model. A table with all the values of the given parameters for each soil class is provided in Doms et al. [2013] and Guillod et al. [2013]. We will provide more information about TERRA_LM and the different soil types in the new version of the manuscript, together with corresponding references.

- *P*. 6, line 30. The experiments are initialized with 50% relative soil moisture, and then the soil moisture is decreased or increased by 25, 50 or 75%. To avoid confusion, please explain what these percentages

mean exactly. For instance: obviously, the initial 50% soil moisture content cannot be reduced by more than 50% if the percentages refer to the same reference soil moisture content. Presumably the initial 50% refers to the volume dictated by the soil water holding capacity, and the 75% refers to "75% of this 50%", but this is not clearly explained.

**We agree with the referee and will provide more detailed information on what the given percentages exactly mean in the new version of the manuscript.**

- *P.* 7, line 16. Why not assigning all points in the domain? Why leaving some points out? Please explain.

**In CORDEX (e.g. Kotlarski et al. [2014]) as well as in calibration studies such as the ones of Bellprat et al. [2012a,b, 2016], when applying a regionalization, not all of the points of the domain are normally considered (the number of considered points is even lower than the one used in our study). In the light of this "modus operandi", we deemed it suitable to apply the given domain subdivision. This choice is also supported by the fact that we actually conducted an analysis for each land point of the domain in the first step of section 2.5 (with similar conclusions). Moreover, the selection of the sub-domains introduces additional uncertainty to the ranking of the different simulations, with likely the same conclusions that could be drawn from other selection choices. Based on the above considerations, we will try to provide more information on the reasons for our choice in the new version of the manuscript.**

- *P.* 8, Figures 3 and 4. I suggest using a different color scheme for Figure 3. Light red for cooling and green for warming is rather unconventional and could be confusing to the readers. In addition, I propose making the difference between Figures 3 and 4 a bit clearer. As I understand it, Figure 3 shows the mean of the anomalies of MH minus PI for experiments with the same parameters sets. Figure 4 then shows the spread around the means shown in Figure 3. Are the spreads more or less normally distributed around the mean?

We agree with the referee on the need to use a different color scheme for Fig.3, as well as making the differences between Fig.3 and Fig.4 clearer in the new version of the manuscript. At the same time, to answer the referee's second question, we include here the same figure for the ensemble spread of summer temperatures MH-PI differences (Fig.4 of the former version of the manuscript), showing in addition a point in correspondence of the grid box where the different ensemble members can be assumed to distribute normally around the mean, according to the results of a Shapiro-Wilk test of normality at a significance level of 0.05 (Fig. 1 of the current document). As you can see, for a large part of the domain the distribution of the ensemble members around the mean can be plausibly approximated with a normal distribution. Despite these results, we want to state here that the information on the distribution of the ensemble members around the mean is not relevant for the purposes of our study. In fact, the analyses presented in Fig. 4 were designed in order to see whether some of the ensemble members differ remarkably from the others, in terms of the differences in mean summer temperatures between the MH and PI periods. For this purpose, we think that Fig. 4 of the former manuscript version, showing the differences between the two most extreme members of the ensemble for each point of the domain, is entirely appropriate. Following the referee's comment, though, we realized that the description of the figure should be revised and that another term (for example simply "maximum absolute differences") should be used in this case instead of the "spread".

- *P*. 9, Figure 7. I wonder what the added value of Figure 7 is. I can see that the simulation with the best performance (nr 2) for PI has a slightly larger deviation from the reference run in MH, and for MH another simulation than nr 2 is closest to the reference run (i.e., simulation 26). In my view, this can be described in the text without showing the figure.

We actually do not agree here with the referee's comment. We think that figure 7 nicely supports the information provided in the text, giving a good example of the behaviour of the ensemble members in the different cases.

[Figure]

Figure 1: **Maximum absolute differences in summer mean temperature MH-PI anomalies, calculated between the different ensemble members. The dots represent the points for which the distribution of the considered ensemble can be approximated by a gaussian distribution, according to the results of a Shapiro-Wilk significance test at a significance level of 0.05.**

- *S*ection 3.2. In this paper, it is suggested that climate models may simulate too warm summer conditions during the MH because of inadequate representations of soil processes and the inability of models to retain soil moisture during summer. Indeed, Figure 5 shows that summer temperatures are reduced when additional soil moisture is artificially added in MH simulations, which would produce a better match with pollen-based reconstructions in Southern Europe. However, what is not discussed here, is that, at the same time, more humid soils decrease the modeling performance in the northern part of Europe, since here the "standard" result with warmer summers during the MH is in good agreement with proxy-based reconstructions. So, with an increase in soil moisture, summer temperatures would also be reduced here, which is definitively not improving the match with proxies. Consequently, in Northern Europe enhanced soil humidity would not provide a solution. This point should be discussed in Section 3.

We want to emphasize again here that the hypothesis that models simulate warmer summer conditions over the Mediterranean region as a consequence of inadequate representations of soil processes and their inability to retain soil moisture is something not proposed in this paper, but in other studies such as the one of Russo and Cubasch [2016] and Bonfils et al. [2004], based on similar considerations on present-day biases of climate models. Here, with our experiments, we simply try to test the plausibility of such hypothesis. As already mentioned, with our results we actually prove that pronounced regional differences in European summer temperatures during the MH might be related to soil-atmosphere interactions: a strong spatial sensitivity of summer near surface temperatures to spring soil moisture perturbation is evident for the considered model. This sensitivity must be taken into account when discussing the results of climate models against proxies: over specific areas, it is more likely that models' results would change as a consequence of a perturbation in spring soil moisture conditions. This means that all the points of the domain for which this sensitivity is high, independently from whether they are in the North or in the South, should be treated carefully in the comparison against proxy-reconstructions. This is true also for the points already showing a good match with proxy-based reconstructions. Following the referee's comment, we will try to better assert the

**latter point in the new version of our manuscript.**

- *S*ection 3. Discussion: I suggest comparing the obtained results here with reports from earlier modelling studies on the MH climate in Europe and to include a discussion of the impact of lateral boundary conditions on the results presented in this study.

**Following the referee's suggestion, we will try to expand our discussion in the different parts of section 3, with a broader consideration of former modeling reports for the MH, whenever possible. Concerning the effect of the boundaries, this point was also highlighted by the first referee and we agree that this is a point requiring more attention throughout the text. The selection of the boundaries plays in fact a very important role for RCMs, being for several domains one of the dominant drivers [Sørland et al., 2021]. Hence, in the new version of the manuscript we will try to include a discussion on the role of the boundaries on the presented results, as suggested by the referee.**

- *P*. 10, Table 4. I suggest omitting Table 4 and simply to describe the information in the text, since the numbers in this table provide little additional value, while the table takes up a lot of space. Besides, the numbers in the table just show the ranking, which does not necessarily provide information on the MAE value of MH versus PI. For example, for T2M, experiment 14 has 2nd rank for MH, but first rank for PI. But this doesn't necessarily mean that the MAE for MH is lower than the MAE for PI, it just means that there is another experiment that has a higher rank for MH (i.e., 29).

**We will consider whether to include table 4 in the supplements, as suggested by the referee. However, concerning the second part of the comment, we want to emphasize that our goal is not to show that the MAE for the same experiment is different in the two periods. Rather, we want to show that the ranking of the experiments based on the MAE changes in the two periods. Therefore, we think that the information provided in the table is very appropriate for the achievement of this goal. Following the referee's comment, we will anyway review the text, trying to make our goals clearer.**

**Technical corrections**

- *P.* 2, line 22. "proxy-reconstructions" should be "proxy-based reconstructions".

**Thanks. We will correct it accordingly.**

- *P.* 4, line 8. Should be "feedbacks" instead of singular feedback.

**We agree and will correct the text accordingly.**

- *P.* 4, line 15. I suggest using "investigating" instead of "supporting"

**We agree and will correct the text accordingly.**

- *P.* 5, line 18: one bracket ")" too many

**Will be corrected.**

- *P.* 5, line 20 should be "is applied"

**Here we are referring to the radiative transfer scheme, to TERRA_LM and to the ice scheme together. Therefore we think that "are applied" is a better choice than the one suggested by the referee.**

- *P.* 6, line 2: should be "A set of parameters is" and "affects" on the next line

**Following the referee's comment we will revise this part of the text, correcting the suggested mistakes.**

- *P. 6, line 12: should be "from one of the reference simulations*

We do not agree here. The PPE is built perturbing the configuration of the reference simulation discussed in the lines from **12** to **22** of page **5**. Following the referee's comment, we will anyway review section **2.3**, trying to make the description clearer whenever possible.

- *P. 6, line 13: should be "is reported" (refers to "a list")*

We agree and will modify the text accordingly.

- *P. 7, line 22. Should be "in both formulas".*

We agree and will modify the text accordingly.

- *P. 14, lines 29 and 35. The DOI's of Jungclaus et al. 2012a, 2012b, 2013 are provided two times.*

Thanks for highlighting this error. We will remove the duplicated DOIs.

- *Figure 2. The colours of number 7 and 8 are not easy to distinguish. I suggest to adjust these colours.*

We agree and will modify the figure following the referee's comment.

- *Figure 4. Caption: should be "mean"*

We agree and will correct the text accordingly.

**References**

O. Bellprat, S. Kotlarski, D. Lüthi, and C. Schär. Objective calibration of regional climate models. *Journal of Geophysical Research: Atmospheres*, 117(D23), 2012a.

O. Bellprat, S. Kotlarski, D. Lüthi, and C. Schär. Exploring perturbed physics ensembles in a regional climate model. *Journal of Climate*, 25 (13):4582–4599, 2012b.

O. Bellprat, S. Kotlarski, D. Lüthi, R. De Elía, A. Frigon, R. Laprise, and C. Schär. Objective calibration of regional climate models: application over europe and north america. *Journal of Climate*, 29(2):819–838, 2016.

F. Boberg and J.H. Christensen. Overestimation of mediterranean summer temperature projections due to model deficiencies. *Nature Climate Change*, 2(6):433–436, 2012.

C Bonfils, N de Noblet-Ducoudré, Joel Guiot, and P Bartlein. Some mechanisms of mid-holocene climate change in europe, inferred from comparing pmip models to data. *Climate Dynamics*, 23(1):79–98, 2004.

D. Carvalho, S. Cardoso Pereira, and A. Rocha. Future surface temperatures over europe according to cmip6 climate projections: an analysis with original and bias-corrected data. *Climatic Change*, 167(1):1–17, 2021.

J. Cattiaux, H. Douville, and Y. Peings. European temperatures in cmip5: origins of present-day biases and future uncertainties. *Climate dynamics*, 41(11-12):2889–2907, 2013.

Jens H Christensen, Fredrik Boberg, Ole B Christensen, and Philippe Lucas-Picher. On the need for bias correction of regional climate change projections of temperature and precipitation. *Geophysical Research Letters*, 35 (20), 2008.

G. Doms, J. Föster, E. Heise, H.J. Herzog, D. Mironov, M. Raschendorfer, T. Reinhardt, B. Ritter, R. Schrodin, J.P. Schulz, and G. Vogel. A description of the nonhydrostatic regional cosmo-model - part ii: Physical parameterizations. Technical report, COSMO - Consortium for Small-Scale Modelling, 2013.

B.P. Guillod, E.L. Davin, C. Kündig, G. Smiatek, and S.I. Seneviratne. Impact of soil map specifications for european climate simulations. *Climate dynamics*, 40(1-2):123–141, 2013.

S. Kotlarski, K. Keuler, O.B. Christensen, A. Colette, M. Déqué, A. Gobiet, K. Goergen, D. Jacob, D. Lüthi, E. Van Meijgaard, G. Nikulin, C. Schär, C. Teichmann, R. Vautard, K. Warrach-Sagi, and V. Wulfmeyer. Regional climate modeling on european scales: a joint standard evaluation of the

euro-cordex rcm ensemble. *Geoscientific Model Development*, 7(4):1297–1333, 2014.

E. Russo and U. Cubasch. Mid-to-late holocene temperature evolution and atmospheric dynamics over europe in regional model simulations. *Climate of the Past*, 12(8):1645–1662, 2016.

S. Samartin, O. Heiri, F. Joos, H. Renssen, J. Franke, S. Brönnimann, and W. Tinner. Warm mediterranean mid-holocene summers inferred from fossil midge assemblages. *Nature geoscience*, 10(3):207–212, 2017.

S.L. Sørland, R. Brogli, P.K. Pothapakula, E. Russo, J. Van de Walle, B. Ahrens, I. Anders, E. Bucchignani, E.L. Davin, M.E. Demory, et al. Cosmo-clm regional climate simulations in the coordinated regional climate downscaling experiment (cordex) framework: a review. *Geoscientific Model Development*, 14(8):5125–5154, 2021.

---

## Author Response (AR2)

Reply to
**2nd Reviewer**
*Russo, E., Fallah, B., Ludwig, P., Karremann, M., and Raible, C.C.:*
*The long-standing dilemma of European summer temperatures at the*
*Mid-Holocene and other considerations on learning from the past for*
*the future using a regional climate model*

Dear referee,

thank you very much again for your time in reviewing our manuscript.

Below we go point by point through your technical corrections, presented in *italic*, detailing how we dealt with your concerns reported in **Bold**.

Sincerely,

Emmanuele Russo

- *P.3, line 4. The "spatial dipole structure" is still not explained. I suggest rephrasing the sentence or to explain what is meant by dipole in this context.*

**We will correct this point in the next version of the manuscript.**

- *Section 2.3. My previous comment: "The text mentions 31 experiments, but Table 3 shows. 30 different experiments." It is still not clear that the 31st experiment is the reference run. I suggest adding this to the caption of Table 4.*

**We will provide this information in the caption of Table 4 in the new version of the manuscript, as suggested by the referee.**

- *Discussion: My previous comment: "I suggest comparing the obtained results here with reports from earlier modelling studies on the MH climate in Europe and to include a discussion of the impact of lateral*

*boundary conditions on the results presented in this study." I could really see where these two important points were discussed, so I propose to include this discussion in the manuscript.*

We suppose that the reviewer wanted to say that he could not really see where these two important points are discussed. Here we have to acknowledge that we actually might not have completely incorporated the changes to the latest version of the manuscript as indicated in our answer to the previous referee's comment. Below we try to motivate our choices, proposing at the same time possible corrections, following the referee's comment.

Concerning the boundaries, we think that we effectively took care of their possible effects on the evinced results by conducting additional sensitivity experiments. Nonetheless, following the referee's comment we realized that some additional information in the text about the relevance of the boundaries for a regional climate model should be provided. However, we believe that it would be more appropriate to provide this information in section 2.4, where the "different boundaries" experiments are already described, instead than in the discussion part as proposed by the referee.

For the other point, we want to acknowledge that a comparison of the obtained results against previous reports from earlier modeling studies of the MH climate, as suggested by the referee, was already conducted in the results and discussion section of the latest version of the manuscript (see p. 8, l. 22-24). Even though such a comparison might be a bit generic, we do believe that it conveys exhaustive information to the reader, relevant for the objectives of the paper. Additionally, considering that several references to former modeling studies of the MH climate are extensively reported throughout different parts of the manuscript we personally decided not to further expand the discussion section on this subject.